# Comparison of Minimally Invasive Surgery with Open Surgery for Type II Endometrial Cancer: An Analysis of the National Cancer Database

**DOI:** 10.3390/healthcare11243122

**Published:** 2023-12-08

**Authors:** Qi Zhang, Michael Silver, Yi-Ju Chen, Jennifer Wolf, Judy Hayek, Ioannis Alagkiozidis

**Affiliations:** 1Department of Gynecologic Oncology, Maimonides Medical Center, Brooklyn, NY 11220, USA; 2Department of Gynecologic Oncology, SUNY Downstate Health Sciences University, Brooklyn, NY 11203, USA; yi-ju.chen@downstate.edu (Y.-J.C.); jennifer.wolf@downstate.edu (J.W.); judy.hayek@downstate.edu (J.H.)

**Keywords:** minimally invasive surgery, type II endometrial cancer, overall survival

## Abstract

Objective: Prior studies comparing minimally invasive surgery with open surgery among patients with endometrial cancer have reported similar survival outcomes and improved perioperative outcomes with minimally invasive surgery (MIS). However, patients with Type II endometrial cancer were underrepresented in these studies. We sought to compare the overall survival and surgical outcomes between open surgery and MIS in a large cohort of women with Type II endometrial cancer. Methods: Using data from the National Cancer Database, we identified a cohort of women who underwent hysterectomy for type II endometrial cancer (serous, clear cell, and carcinosarcoma) between January 2010 and December 2014. The primary outcome was a comparison of the overall survival for MIS with that for the open approach. The secondary outcomes included a comparison of the length of hospital stay, readmission within 30 days of discharge, and 30- and 90-day mortality. Outcomes were compared between the cohorts using the Mann–Whitney U test, Pearson’s chi-square test, or Fisher’s exact test. Multivariable logistic regression with inverse propensity weighting was used to determine clinical characteristics that were statistically significant predictors of outcomes. *p* values < 0.05 were considered significant. Results: We identified 12,905 patients with Type II, Stage I–III endometrial cancer that underwent a hysterectomy. In total, 7123 of these women (55.2%) underwent MIS. The rate of MIS increased from 39% to 64% over four years. Women who underwent MIS were more often White, privately insured, older, and had a higher income. The laparotomy group had a higher rate of carcinosarcoma histology (30.9% vs. 23.6%, *p* < 0.001), stage III disease (38.4% vs. 27.4%, *p* < 0.001), and larger primary tumors (59 vs. 45 mm, *p* < 0.001). Lymph node dissection was more commonly performed in the MIS group (89.6% vs. 85.4%, *p* < 0.001). With regard to adjuvant therapy, subjection to postoperative radiation was more common in the MIS group (37% vs. 40.1%, *p* < 0.001), while chemotherapy was more common in the laparotomy group (37.6% vs. 33.9%, *p* < 0.001). The time interval between surgery and the initiation of chemotherapy was shorter in the MIS group (39 vs. 42 days, *p* < 0.001). According to the results of propensity-score-weighted analysis, MIS was associated with superior overall survival (101.7 vs. 86.7 months, *p* = 0.0003 determined using the long-rank test), which corresponded to a 10% decreased risk of all-cause mortality (HR 0.9; CI 0.857–0.954, *p* = 0.0002). The survival benefit was uniform across all three histology types and stages. MIS was associated with superior perioperative outcomes, including shorter length of stay (1 vs. 4 days, *p* < 0.001), lower 30-day readmission rates (2.5% vs. 5%), and lower 30- and 90-day postoperative mortality (0.5% vs. 1.3% and 1.5% vs. 3.6%, respectively; *p* < 0.001 for both). The increased adoption of MIS from 2010 to 2014 corresponds to a decrease in 90-day postoperative mortality (2.8% to 2.2%, r = −0.89; *p* = 0.04) and overall mortality (51% to 38%, r = −0.95; *p* = 0.006). Conclusions: In a large cohort of patients from the National Cancer Database, MIS was associated with improved overall survival and superior perioperative outcomes compared to open surgery among women with Type II endometrial cancer. A decrease in postoperative mortality and a shorter interval between surgery and the initiation of chemotherapy may contribute to the survival benefit of MIS. A racial and economic disparity in the surgical management of Type II endometrial cancer was identified, and further investigation is warranted to narrow this gap and improve patient outcomes.

## 1. Introduction

Endometrial cancer is the most common gynecologic malignancy in the United States, with more than 65,000 new cases per year. In a landmark study published in 1983, Bokhman et al. described two distinct types of endometrial cancer based on histologic and molecular characteristics [1]. The endometrioid type (Type I) constitutes 80–90% of all sporadic endometrial cancers, while the non-endometrioid type (Type II) accounts for the remaining 10–20%. While the incidence of Type II tumors is low compared to that of Type I, excess mortality is associated with Type II EC. In an analysis of Surveillance, Epidemiology, and End Results (SEER) data, Hamilton et al. reported that while 11% of ECs were Type II, 47% of deaths corresponded to this subtype, and stage-adjusted 5-year overall survival rates for Type II tumors are significantly worse compared to Type I tumors [2].

Endometrial cancer type II is surgically staged with hysterectomy, bilateral salpingo-oophorectomy, and lymph node evaluation. Laparoscopy has become the standard surgical approach for patients with early-stage uterine carcinoma due to the results of studies such as LAP2, which demonstrated that there was no negative effect of the MIS approach on oncologic outcomes [3]. Another randomized trial (Laparoscopic Approach to Cancer of the Endometrium, LACE) showed no difference in overall and recurrence-free survival between the two surgical approaches [4]. Both studies established the superiority of the MIS approach regarding perioperative outcomes. However, the results of these studies may not apply to patients with aggressive histology types. Type II tumors were poorly represented in LAP2, accounting for only 20% of the enrolled patients, while LACE included only patients with endometrioid adenocarcinoma. 

In 2018, the Laparoscopic Approach to Cervical Cancer (LACC) trial reported worse oncologic outcomes for patients undergoing minimally invasive surgery for cervical cancer [5]. Several factors have been linked to this unexpected finding, encompassing tumor dissemination influenced by surgical-technique-related factors, the potential impact of pneumoperitoneum on tumor seeding and the immune system, and the aggressive nature of cervical cancer and its propensity to subvert a patient’s immune response and primarily grow through VEGF activation and angiogenesis [6,7,8]. Type II endometrial cancer exhibits more aggressive behavior and has a higher metastatic potential compared to Type I, posing a significant risk of intraperitoneal and lymphatic recurrence. The findings of the LACC trial have raised concerns about the oncologic safety of MIS among patients with Type II endometrial cancers. Furthermore, an association of worse disease-free survival among patients with intermediate-risk endometrial cancer who underwent robotically assisted laparoscopy as opposed to open surgery was recently reported by Song et al. [9]. In the widest single-institution retrospective study comparing MIS with open surgery in Type II endometrial cancer published to date, Monterossi et al. found no difference in survival outcomes in Stages I and II and a trend of worse overall survival with MIS in stage III [10]. In a systematic review of the literature that included nine retrospective studies, it was noted that MIS appears to offer better perioperative and postoperative outcomes and comparable oncological outcomes [11]. The low incidence of Type II endometrial tumors makes this disease difficult to study in a prospective, randomized manner, and a large epidemiologic study based on a national cancer registry would be the best next alternative. 

We designed a study to evaluate survival and surgical outcomes using a large cohort of patients from the National Cancer Database. The primary objective was to compare overall survival between patients with Type II endometrial cancer who underwent minimally invasive and those who underwent an open hysterectomy. A secondary objective was to compare surgical outcomes such as length of stay, readmission rates, and postoperative mortality.

## 2. Materials and Methods

Data were obtained from the National Cancer Database, which includes data on patients who received some element of their cancer care (treatment or diagnosis) at a cancer program accredited by the Commission on Cancer-Accredited Centers. Data cover more than 70% of newly diagnosed cases collected from approximately 1500 facilities. We identified a cohort of women who underwent hysterectomy as the primary treatment for Type II endometrial cancer (serous, clear-cell, and carcinosarcoma) between January 2010 and December 2014. We excluded patients who did not have a hysterectomy as primary treatment, those whose primary treatment was unknown, those with stage IV disease, and those for whom there was a lack of pathological confirmation of cancer. Given the aggressive histology of the cases included in the study and the absence of information regarding the molecular subtype, in accordance with the revised 2023 FIGO staging system, the cases were classified into stages IC, IIC, and III [12]. The surgical approach documented included open or minimally invasive surgery (either laparoscopic or robotically assisted). Our analysis was based on the intention-to-treat model, in which we included any cases initiated as minimally invasive surgery even if they were later converted to open surgery. We compared the two groups in terms of age, race, co-morbidities, stage, histology type, primary tumor size, and adjuvant therapy. Comorbid conditions were analyzed using the Charlson/Deyo Score provided by the National Cancer Database. The Charlson/Deyo value is a weighted score derived from the sum of the scores for each of the comorbid conditions listed in the Charlson Comorbidity Score; a score of 0 indicates “no comorbid conditions recorded”. Histology in the NCDB PUF dictionary was reported as ICD-O-3 codes reported by SEER registries.

The primary outcome was a comparison of overall survival between patients undergoing minimally invasive surgery (MIS) and those being treated with open surgical management. Secondary endpoints included length of hospital stay, readmission within 30 days of discharge, and 30- and 90-day mortality. 

## 3. Statistical Analysis

Patient demographics, clinical scores, treatment, and tumor characteristics were analyzed using Pearson’s chi-square or Fisher’s exact test for categorical data and Mann–Whitney U for continuous data. 

A propensity score was calculated to control for confounding using multivariable logistic regression with nine potential confounding variables (Year of diagnosis, Age, Race, Histology, Analytic Stage Group, Chemotherapy, Radiation Therapy, Lymph nodes dissected, and Tumor Size). We then applied inverse propensity weighting to create a pseudo-population to balance the measured confounding variables. To reduce variability in the weighted sample, we applied stabilized weights, where traditional inverse propensity score weighting applies a weight of 1/(propensity score), where group = MIS and 1/(1 − propensity score), wherein group = Open; the stabilized weights were applied as follows: group = MIS; (probability of receiving MIS)/(propensity score), where group = Open; (1 − probability of receiving MIS)/(1 − propensity score).

For the Inverse probability of treatment weighted (IPTW) sample, we compared groups using weighted binary logistic regression for all patient characteristics, clinical scores, tumor characteristics, treatment, and outcomes.

Rates of 30-day mortality, 30-day readmission, and 90-day mortality were plotted over time using both raw percentages and IPTW percentages.

Median survival was estimated using the product-limit method, plotted with a Kaplan–Meier curve, and compared with a log-rank test. This analysis was conducted for both the raw and IPTW samples.

Finally, multivariable Cox proportional hazards models were created to estimate the effect of MIS on the hazards function while also controlling for relevant covariates. A weighted Cox model was used to estimate the same effect for the IPTW sample. These results were compared as a sensitivity analysis to demonstrate the robustness of the findings.

A two-tailed *p*-value of less than 0.05 was considered statistically significant. Analyses were performed using SPSS Version 28 (IBM Corp, Armonk, NY, USA).

## 4. Results

### 4.1. Patient Characteristics

Between January 2010 and December 2014, we identified 12,905 patients that underwent a hysterectomy for Type II uterine cancer. A total of 7123 (55%) underwent minimally invasive surgery. The rate of MIS increased from 39% in 2010 to 64% in 2014. Women who underwent minimally invasive hysterectomy were more likely to be White and privately insured and have a higher income. The mean ages for both groups were relatively similar (67 versus 68). Carcinosarcoma histology was more common in the open group (30.7% vs. 23.4%). In the open group, patients were more likely to have stage III disease (38.4% vs. 27.4%) and larger primary tumors. Postoperative radiation treatment was more common in the MIS group (40.1% vs. 37%), while chemotherapy was more common in the open group (37.6% vs. 33.9%) (Table 1). 

### 4.2. Survival Analysis

The median follow-up was 54 months in the open group and 59 months in the MIS group. The multivariate analysis revealed that the demographic factors associated with worse OS included older age (HR 1.035, *p* < 0.001) and African American race (HR 1.28, *p* < 0.001). The pathologic factors associated with increased mortality were stage, histology type, and size of the primary tumor. Stage IIC and III were associated with worse OS (HR 1.8 and 3 respectively, *p* < 0.001). Serous and carcinosarcoma histologies were associated with worse OS compared to clear-cell histology (HR 1.2 and 1.7 respectively, *p* < 0.001). A larger size of the primary tumor was also associated with decreased OS. Evaluation of the adjuvant therapy modalities showed that both chemotherapy and radiation were associated with improved survival (HR 0.8 for both, *p* < 0.001). 

With regard to the surgical approach, MIS was associated with better OS (HR 0.9, *p* < 0.001) (Table 2). In propensity-score-weighted analyses, MIS was associated with superior overall survival (102 vs. 87 months, *p* = 0.003 determined via the long-rank test), which corresponds to a 10% lower risk of death from any cause (HR 0.9; CI 0.857–0.954, *p* = 0.0002). Weighted survival functions for the MIS group and the open group are plotted in Figure 1. MIS was associated with better survival outcomes across all three histology types and stages (Figure 2). The increased adoption of MIS from 2010 to 2014 corresponds to a decrease in overall mortality (51% to 38% at 12 months, r = −0.95; *p* = 0.006) (Figure 3).

### 4.3. Perioperative Outcomes

MIS was associated with superior perioperative outcomes. The mean length of stay was shorter in the MIS group (1 vs. 4 days with *p*-value < 0.001). The 30-day readmission rate was higher in the open group (5% vs. 2.5% with *p*-value < 0.001). Furthermore, MIS was associated with lower 30 and 90-day postoperative mortality (0.5% vs. 1.3% and 1.5% vs. 3.6%, respectively) (*p*-value < 0.001) (Table 1). The increase in the use of MIS between 2010 and 2014 corresponds to a decrease in postoperative death rates (Figure 3). The time interval from surgery to chemotherapy was shorter for the MIS group (38 vs. 41 days, *p* = 0.013) (Table 1).

## 5. Discussion

Our findings suggest that minimally invasive surgery is associated with better overall survival compared to open hysterectomy among patients with Type II uterine cancer. MIS was also associated with improved perioperative outcomes such as a shorter length of hospital stay and decreased readmission and 30- and 90-day mortality rates. Prior studies have demonstrated superior perioperative outcomes with MIS without a detrimental effect on the oncologic outcomes. However, this is the first study that demonstrates the survival benefit of MIS for individuals with Type II uterine cancer.

LAP-2 was a landmark study that established the oncologic safety of MIS for patients with endometrial cancer. In a post hoc analysis of LAP2 patients with uterine serous, clear-cell carcinosarcoma and Grade III endometrial adenocarcinoma, there was no difference in progression-free and overall survival between MIS and open surgical approach [13]. In the largest systematic review published to date, MIS was associated with improved perioperative outcomes and similar oncologic outcomes compared to open surgery [14]. 

To our knowledge, the present study offers the largest analysis to date on the use of MIS in treating Type II uterine cancer. It is likely that the larger number of patients included in this study compared to that of prior studies provided sufficient power for the detection of the survival benefit offered by MIS. Furthermore, we assessed several potential confounding factors between the study groups, such as histology type, stage, tumor size, and adjuvant therapy. According to our data, MIS is an independent prognostic factor of improved survival among patients with Type II uterine cancer.

Several factors may have contributed to our study’s superior survival outcomes with MIS. First, MIS is associated with decreased postoperative mortality rates at 30 and 90 days, affecting the overall survival of the study cohort. Second, the distinct biology of Type II uterine cancer may make this disease suitable for treatment using an MIS approach. In 2018, both the LACC trial and a large epidemiologic study utilizing the NCDB reported poorer oncologic outcomes among patients who underwent minimally invasive radical hysterectomy for cervical cancer [5,15]. Etiologic factors that have been implicated in the worse oncologic outcomes attained using MIS in treating cervical cancer include the increased propensity for tumor spillage due to the uterine manipulator [5,14] or an effect of CO_2_ on tumor cell spread. Beyond the increased risk of recurrence, the recurrence pattern diverges notably following minimally invasive radical hysterectomy for cervical cancer compared to open surgery. Specifically, minimally invasive radical hysterectomy exhibits a statistically significant association with a higher risk of peritoneal carcinomatosis [16]. Microscopic peritoneal spread at the time of surgery can lead to subsequent growth since most cervical cancer patients do not receive adjuvant therapy after a radical hysterectomy. In contrast, most patients with Type II endometrial cancer receive adjuvant therapy, and chemotherapy has a role in managing all stages of the disease. Adjuvant therapy targets microscopic disease in the immediate postoperative setting, and treatment delays have been associated with worse outcomes for various malignancies [17,18]. A possible explanation for the superior survival outcomes with MIS in oncologic surgery is that the superior perioperative outcomes with MIS allow faster recovery and earlier initiation of adjuvant therapy, leading to superior survival outcomes. Interestingly, the time interval from surgery to chemotherapy was shorter for the MIS cohort in our study. However, the marginal difference in the time taken to initiate chemotherapy is small, suggesting that it is unlikely to be the sole factor accounting for the observed survival advantage bestowed by MIS. Furthermore, an association between surgical stress and immunosuppression has been well described in the literature. The use of MIS could likely decrease surgical stress and the negative impact of surgery on the antitumor immune response [19,20,21,22].

We acknowledge several important limitations of our study. Although the NCDB represents women from a large number of hospitals, these data may not be representative of the entire population. Another significant limitation of this study is the absence of information about recurrence, subsequent treatment at the time of recurrence, and cause of death recorded in the National Cancer Database. Furthermore, there was no information on factors affecting the choice between open surgery or the MIS approach that could have led to potential selection bias. It may be suggested that high-risk patients (based on preoperative pathology or imaging) were selected to undergo open surgery. To eliminate the effect of confounding factors, a propensity-score-weighted analysis was performed, demonstrating that MIS is independently associated with improved survival. An analysis of the trends in survival in association with the rates of MIS over time reinforced the conclusions drawn by the multivariable model. Finally, operative morbidity is a significant factor that affects the choice of surgical approach, and our data lack information on this aspect.

Another limitation of our study pertains to the absence of tumor molecular classification data within the National Cancer Database. The histologic classification of endometrial cancer used in this study is prone to misdiagnosis, with studies indicating approximately 30% uncertainty among pathologists [23,24,25]. Molecular classification has become a valuable tool for prognosis assessment and treatment guidance in recent years. Endometrial cancer has been categorized into four molecular subgroups (POLE ultramutated, MSI-H, copy number high, and copy number low), each with a distinct prognosis and response to treatment. Investigating the connection between tumor molecular classification and the surgical approach employed is a potential topic for future studies [26].

A significant finding of our study is that racial and social disparities in the management of endometrial cancer extend to its surgical care. The healthcare disparities in endometrial cancer have been extensively documented in the literature, stemming from various factors [27,28,29,30]. These encompass biological dissimilarities, genetic variations, socioeconomic disparities, cultural influences affecting healthcare-seeking behaviors, implicit biases within healthcare systems, and unequal access to high-quality care and clinical trials [31,32,33,34,35]. Consistent with previous research, African American women exhibited poorer overall survival rates in our study. Additionally, while minimally invasive surgery (MIS) is associated with improved survival and perioperative outcomes, its accessibility remains unequal among different racial and social groups. Our study indicates that women undergoing minimally invasive hysterectomy were more likely to be White, be privately insured, and possess higher income levels. These findings underscore the disparities in the provision of surgical care, potentially contributing to divergent survival outcomes. Efforts centered on mitigating racial disparities in uterine cancer necessitate a multifaceted approach. This encompasses promoting equitable access to healthcare services, enhancing health education and awareness within marginalized communities, diversifying clinical trials to include diverse populations, implementing culturally competent care, and addressing systemic barriers in healthcare delivery. Ensuring the availability of all surgical modalities and specialized surgeons within marginalized communities could significantly enhance survival outcomes.

In conclusion, our study includes a large cohort of patients with Type II uterine cancer derived from a nationwide, multicenter database. Based on our multivariate propensity-score-weighted analysis, MIS is associated with improved overall survival compared to open hysterectomy and offers improved postoperative outcomes. Based on this large data set, MIS, when feasible, should be the recommended surgical approach for patients with Type II uterine cancer.

## Figures and Tables

**Figure 1 healthcare-11-03122-f001:**
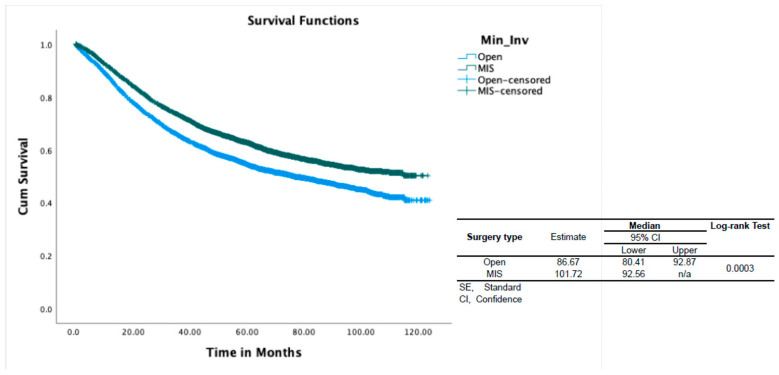
Overall survival of patients with Type II endometrial cancer according to surgical approach. CI, confidence interval; SE, standard error; n/a, not applicable.

**Figure 2 healthcare-11-03122-f002:**
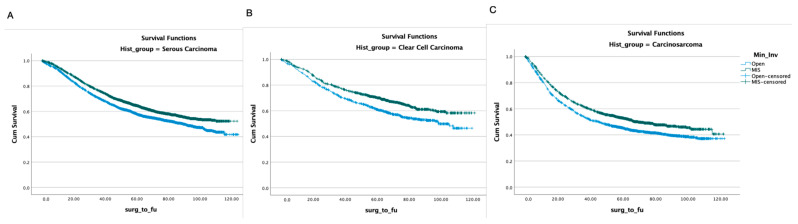
Overall survival according to surgical approach in different histology types. (**A**) serous carcinoma, (**B**) clear-cell carcinoma, and (**C**) carcinosarcoma.

**Figure 3 healthcare-11-03122-f003:**
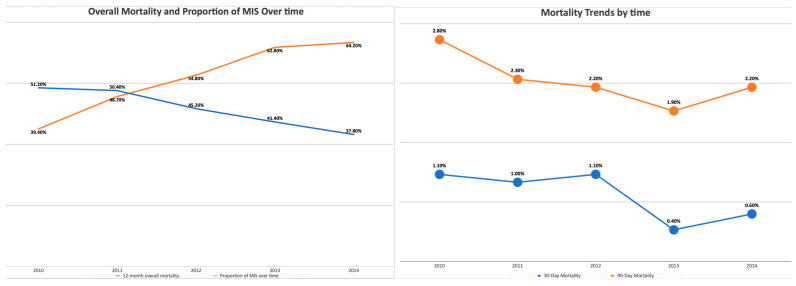
The increased adoption of MIS from 2010 to 2014 corresponds to a decrease in postoperative mortality and overall mortality.

**Table 1 healthcare-11-03122-t001:** Patient demographics and disease characteristics (unweighted and after IPTW).

		Unweighted			After IPTW	
Variables		Surgical Type			Surgical Type	
	Open (n = 5782)	MIS (n = 7123)	*p*-value	Open (n = 5578)	MIS (n = 6867)	*p*-value
Follow-up in months Median(IQR)	55.9 (23–78.9)	57.2 (26–74.6)				
Age	67 (61–74)	68 (62–74)	<0.001	68 (62–74)	67 (62–74)	0.965
Race, n (%)						
White	3967 (68.1)	5450 (75.7)		4037 (72.4)	4970 (72.4)	0.985
Black	1559 (26.8)	1378 (19.1)		1257 (22.5)	1546 (22.5)	
Asian/Pacific Islander	178 (3.1)	218 (3)	<0.001	168 (3)	207 (3)	
Other	63 (1.1)	78 (1.1)		61 (1.1)	74 (1.1)	
Unknown	60 (1)	73 (1)		56 (1)	69 (1)	
Ethnicity, n (%)						
Non-Hispanic	5277 (90.6)	6574 (91.3)		5056 (90.6)	6276 (91.4)	0.13
Hispanic	367 (6.3)	444 (6.2)	0.072	358 (6.4)	417 (6.1)	
Unknown	183 (3.1)	179 (2.5)		164 (2.9)	174 (2.5)	
Insurance, n (%)						
No	219 (3.8)	141 (2)	<0.001	194 (3.5)	147 (2.1)	0.001
Private	1968 (33.8)	2541 (35.3)		1836 (32.9)	2478 (36.1)	
Medicaid/Medicare/Other Public	3541 (60.8)	4456 (61.9)		3463 (62.1)	4187 (61)	
Unknown	99 (1.7)	59 (0.8)		85 (1.5)	55 (0.8)	
Median Income Quartiles, n (%)						
<USD 30,000	885 (17)	840 (13.1)	<0.001	794 (16)	860 (14.1)	<0.001
USD 30,000–USD 34,999	934 (17.9)	1007 (15.7)		860 (17.3)	967 (15.8)	
USD 35,000–USD 45,999	1381 (26.5)	1721 (26.9)		1330 (26.7)	1642 (26.9)	
USD 46,000	2018 (38.7)	2826 (44.2)		1992 (40)	2647 (43.3)	
Charlson Comorbidity Index, n (%)						
0	4117 (70.7)	5200 (72.3%)	0.164	3949 (70.8)	4935 (71.9)	0.162
1	1336 (22.9)	1584 (22%)		1269 (22.7)	1522 (22.2)	
2	293 (5)	317 (4.4%)		281 (5)	317 (4.6)	
3	81 (1.4)	96 (1.3%)		79 (1.4)	93 (1.4)	
Year of diagnosis, n (%)						
2010	1267 (21.7)	823 (11.4%)	<0.001	1217 (21.8)	790 (11.5)	<0.001
2011	1228 (21.1)	1163 (16.2%)		1179 (21.1)	1119 (16.3)	
2012	1140 (19.6)	1383 (19.2%)		1088 (19.5)	1317 (19.2)	
2013	1064 (18.3)	1811 (25.2%)		1023 (18.3)	1724 (25.1)	
2014	1128 (19.4)	2017 (28%)		1070 (19.2)	1917 (27.9)	
Histology, n (%)						
Serous Carcinoma	3288 (52.9)	4539 (63.7%)	<0.001	3395 (60.9)	4182 (60.9)	0.958
Clear-Cell Carcinoma	708 (12.2)	901 (12.6%)		694 (12.4)	855 (12.5)	
Carcinosarcoma	1786 (30.9)	1683 (23.6%)		1489 (26.7)	1830 (26.6)	
Lymph Node Dissection	4937 (85.4%)	6380 (89.6%)	<0.001	4767 (85.5)	6139 (89.4)	<0.001
Tumor Size (mm)	59 (32–120)	45 (27–92)	<0.001	55 (30–115)	48 (29–92)	0.99
FIGO stage, n (%)						
IC	2988 (51.7%)	4646 (65.2%)	<0.001	3318 (59.5)	4085 (59.5)	0.991
IIC	571 (9.9%)	525 (7.4%)		473 (8.5)	581 (8.5)	
III	2223 (38.4%)	1952 (27.4%)		1786 (32)	2201 (32)	
Chemotherapy, n (%)	2156 (37.6%)	2407 (33.9%)	<0.001	2004 (35.9)	2461 (35.8)	0.912
Days from surgery to chemotherapy	41 (30–56)	38 (29–51)	<0.001	41 (30–56)	38 (29–51)	<0.001
Radiation therapy, n (%)	2101 (37%)	2813 (40.1%)	<0.001	2191 (39.3)	2697 (39.3)	0.992
Length of Hospital Stay	4 (3–5)	1 (1–2)	<0.001	4 (3–5)	1 (1–2)	<0.001
Months from Surgery to Last visit	53.55 (21.4–77.48)	58.63 (29.21–75.55)	<0.001	55.94 (23.57–78.94)	57.22 (26.07–74.63)	0.805
Death, n (%)	2839 (48.7%)	2902 (40.3%)	<0.001	2607 (46.7)	2932 (42.7)	<0.001
30 Day Readmission, n (%)	289 (5.0%)	177 (2.5%)		266 (4.8)	175 (2.6)	<0.001
30 Day Mortality, n (%)	76 (1.3%)	37 (0.5%)	<0.001	64 (1.2)	37 (0.5)	<0.001
90 Day Mortality, n (%)	207 (3.6%)	109 (1.5%)	<0.001	165 (3)	112 (1.6)	<0.001

MIS, minimally invasive surgery. FIGO, The International Federation of Gynecology and Obstetrics.

**Table 2 healthcare-11-03122-t002:** Multivariable overall survival analysis of patients with Type II EC.

	HR	95% CI		Sig.
Variables		Lower	Upper	
Age	1.032	1.029	1.035	<0.001
Race				
White	ref.	ref.	ref.	ref.
Black	1.261	1.184	1.343	<0.001
Asian/Pacific Islander	0.871	0.73	1.038	0.123
Other	1.067	0.819	1.39	0.63
Unknown	1.126	0.861	1.474	0.386
Histology				
Serous carcinoma	ref.	ref.	ref.	ref.
Clear-cell carcinoma	0.886	0.81	0.969	0.008
Carcinosarcoma	1.53	1.442	1.624	<0.001
FIGO stage				
IC	ref.	ref.	ref.	ref.
IIC	1.775	1.609	1.957	<0.001
III	3.107	2.933	3.291	<0.001
MIS	0.905	0.857	0.956	<0.001
Chemotherapy	0.77	0.725	0.817	<0.001
Radiation therapy	0.801	0.756	0.848	<0.001
Lymph node dissection	0.586	0.545	0.631	<0.001
Tumor Size (10 mm increment)	0.998	0.998	0.999	<0.001

Multivariable Cox regression analysis of baseline clinical covariates and effects on OS for patients with Type II endometrial cancer. The HR, CI, and *p*-value are presented for each characteristic assessed. HR, hazard ratio. CI, confidence interval.

## Data Availability

The data are available upon request.

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
