# Peer review of "Comparison of Minimally Invasive Surgery with Open Surgery for Type II Endometrial Cancer: An Analysis of the National Cancer Database"

_healthcare, 2023, doi:10.3390/healthcare11243122_

Round 1

Reviewer 1 Report

Comments and Suggestions for Authors

The study is well conducted, with well-organized, clear and well-explained analysis. Unfortunately, however, I must underline that, in light of the new classification of endometrial carcinoma, this classification and therefore this manuscript completely lose scientific meaning, therefore, while appreciating the authors' work, it finds no scientific reason.

As reported in this article "Berek JS, Matias-Guiu : 2023. Int J Gynaecol Obstet. 2023 Aug;162(2):383-394. doi: 10.1002/ijgo.14923. Epub 2023 Jun 20. Erratum in: Int J Gynaecol Obstet. 2023 Oct 6;: PMID: 37337978. " Histological tumor type is an important prognostic predictor in endometrial carcinoma. All endometrial carcinomas should be classified according to the 5th edition of WHO Classification of Tumors, Female Genital Tumors.1 The following different histological types have been recognized: (1) endometrioid carcinoma (EEC), of low grade (grades 1 and 2) or high grade (grade 3); (2) serous carcinoma (SC); (3) clear cell carcinoma (CCC); (4) mixed carcinoma (MC); (5) undifferenti-ated carcinoma (UC); (6) carcinosarcoma (CS); (7) other unusual types, such as mesonephric-like; and (8) gastrointestinal mucinous type carcinomas. These different histological types have different molecular features, microscopic appearance, precursor lesions, and natural history. Previous studies have shown that histological typing may be essential in staging.2 In this revised FIGO staging, non-aggressive histological types are composed of low-grade (grades 1 and 2) EECs, while aggressive histological types are composed of high-grade EECs (grade 3), SC, CCC, MC, UC, CS, and mesonephric-like and gastro-intestinal type mucinous carcinomas.
This new classification invalidates the content of the manuscript, which divides endometrial carcinomas according to the old classification into two classes. This means that the histopathological assumptions on which the entire manuscript is based are no longer valid to date, although the discussion and the results are undoubtedly conducted in a valid and correct manner.

Author Response

Responses to reviewer #1

General comment by the reviewer:

The study is well conducted, with well-organized, clear and well-explained analysis. Unfortunately, however, I must underline that, in light of the new classification of endometrial carcinoma, this classification and therefore this manuscript completely lose scientific meaning, therefore, while appreciating the authors' work, it finds no scientific reason.

As reported in this article "Berek JS, Matias-Guiu : 2023. Int J Gynaecol Obstet. 2023 Aug;162(2):383-394. doi: 10.1002/ijgo.14923. Epub 2023 Jun 20. Erratum in: Int J Gynaecol Obstet. 2023 Oct 6;: PMID: 37337978. " Histological tumor type is an important prognostic predictor in endometrial carcinoma. All endometrial carcinomas should be classified according to the 5th edition of WHO Classification of Tumors, Female Genital Tumors.1 The following different histological types have been recognized: (1) endometrioid carcinoma (EEC), of low grade (grades 1 and 2) or high grade (grade 3); (2) serous carcinoma (SC); (3) clear cell carcinoma (CCC); (4) mixed carcinoma (MC); (5) undifferenti-ated carcinoma (UC); (6) carcinosarcoma (CS); (7) other unusual types, such as mesonephric-like; and (8) gastrointestinal mucinous type carcinomas. These different histological types have different molecular features, microscopic appearance, precursor lesions, and natural history. Previous studies have shown that histological typing may be essential in staging.2 In this revised FIGO staging, non-aggressive histological types are composed of low-grade (grades 1 and 2) EECs, while aggressive histological types are composed of high-grade EECs (grade 3), SC, CCC, MC, UC, CS, and mesonephric-like and gastro-intestinal type mucinous carcinomas.

This new classification invalidates the content of the manuscript, which divides endometrial carcinomas according to the old classification into two classes. This means that the histopathological assumptions on which the entire manuscript is based are no longer valid to date, although the discussion and the results are undoubtedly conducted in a valid and correct manner.

-Author response

We greatly appreciate the reviewer's positive feedback regarding the study's quality. The observation regarding the update in the FIGO staging system is duly noted. Taking into account the aggressive histology type of the included cases and the absence of information on molecular classification, the stage assignment based on the updated system would be IC, IIC, and III. This adjustment has been made in the manuscript and reflected in the tables

Reviewer 2 Report

Comments and Suggestions for Authors

This interesting paper analyses a large database of type II endometrial cancer patients to verify the efficacy and risks associated with minimally invasive surgery. The topic is of value after the LACC trial that identified minimally invasive surgery as a potential risk for overall survival in cervical cancer patients.

Some revisions are required:

1.      Line 57: the sentence over-simplified the results of the LACC trial; the supposed reasons for the impaired results in cervical cancer patients are different.

2.      Table 1: please specify that the second group of patients is the propensity score group (in the Table and in the legend)

3.      Line 150: the lower survival of African American patients is statistically significative. Do you think that different follow-up and different regimens in case of relapse in this group of patients can explain lower survival? The higher proportion of laparotomic surgery in these patients could be a bias. Please comment

4.      Line 164:  the sentence should be modified; 51% and 38% are related to the adoption of MIS and not to the survival

5.      Figure 3: Overall mortality and Proportion of MIS over time graph does not present survival figures on the vertical axis. Please insert

6.      Figure 3: Mortality trends by time. The impressive decrease of mortality in 2013 could be linked to the rise of MIS adoption in the previous years. Please comment on this

7.      Line 221: earlier initiation of adjuvant therapy of 3 days is hardly to be considered as fundamental for superior survival. Please modify.

8.      Bibliography has to be fully reviewed as most of the references are not in line  with the style of the journal.

Author Response

Responses to Reviewer #2:

General comment by the reviewer:

This interesting paper analyses a large database of type II endometrial cancer patients to verify the efficacy and risks associated with minimally invasive surgery. The topic is of value after the LACC trial that identified minimally invasive surgery as a potential risk for overall survival in cervical cancer patients.

-Author response

We thank the reviewer for his valuable and very constructive comments.

Specific comments:

Reviewer comment/question 1:

Line 57: The sentence over-simplified the results of the LACC trial; the supposed reasons for the impaired results in cervical cancer patients are different.

Author response 1:

We concur with the reviewer's assessment that the underlying cause behind the unexpected outcomes in the LACC trial remains unclear. Several factors have been suggested, encompassing potential tumor dissemination owing to the surgical technique, the impact of pneumoperitoneum on the immune system, and the unique tumor biology associated with cervical cancer. Lines: 57-61

Reviewer comment/question 2:

 Table 1: please specify that the second group of patients is the propensity score group (in the Table and in the legend)

Author response 2:

Modified

Reviewer comment/question 3:

Line 150: the lower survival of African American patients is statistically significative. Do you think that different follow-up and different regimens in case of relapse in this group of patients can explain lower survival? The higher proportion of laparotomic surgery in these patients could be a bias. Please comment

Author response 3:

The observed lower overall survival among African American patients is statistically significant, aligning with previous literature emphasizing the survival disparity in this demographic regarding endometrial cancer. Contributing factors may encompass distinct tumor biology and socioeconomic elements. Our study underscores that this healthcare disparity persists within the realm of surgical management for endometrial cancer. Lines:254-273

Reviewer comment/question 4:

 Line 164:  the sentence should be modified; 51% and 38% are related to the adoption of MIS and not to the survival

Author response 4:

We would like to clarify that the mortality rates of 51% and 38% represent the overall mortality among patients diagnosed during the initial and final year of data collection, respectively. The decline, marked by a statistically significant difference with a p-value of 0.006, coincides with the increased adoption of minimally invasive surgery (MIS) during the same period.

Reviewer comment/question 5:

Figure 3: Overall mortality and Proportion of MIS over time graph does not present survival figures on the vertical axis. Please insert

Author response 5:

We would like to clarify that the vertical axis presents mortality rates. Overall mortality on the left and postoperative mortality (30 and 90-day) on the right.

Reviewer comment/question 6:

Figure 3: Mortality trends by time. The impressive decrease of mortality in 2013 could be linked to the rise of MIS adoption in the previous years. Please comment on this

Author response 6:

We acknowledge the correlation between the reduction in overall mortality and the concurrent rise in the utilization of minimally invasive surgery (MIS) during this period. This temporal trend strongly suggests the potential survival advantages associated with MIS.

Reviewer comment/question 7:

Line 221: earlier initiation of adjuvant therapy of 3 days is hardly to be considered as fundamental for superior survival. Please modify.

Author response 7:

We concur that the marginal difference in the time to initiate chemotherapy is small, suggesting it is unlikely to be the sole factor accounting for the observed survival advantage of MIS.  Lines: 224-229

Reviewer comment/question 8:

Bibliography has to be fully reviewed as most of the references are not in line with the style of the journal.

Author response 8:

Corrected

Reviewer 3 Report

Comments and Suggestions for Authors

Dear Editor,

this study, which compares as the main outcome overall survival after MIS and laparotomy operations in women with endometrial carcinoma type II, has two main limitations - as the authors write in the discussion section. First, it is not clear from the database whether the women died as a result of their oncological diseases or other, for example, internal diseases. Second, both groups of women may be very heterogeneous, subjects were not randomized, and nothing is known about their comorbidities, ASA, or performance status. It is possible that patients who were more obese and more ill were selected for laparotomy, and therefore died earlier than women in the MIS group. I leave it to the editor to decide whether to accept the article for publication.

Another comment

 Lines 44-45

Endometrial cancer is surgically staged with hysterectomy, bilateral salpingo-oophorectomy, and lymph node evaluation.

Note: The better statement is: “Endometrial cancer type II is surgically staged with hysterectomy, bilateral salpingo-oophorectomy, and lymph node evaluation.“

Low risk endometrial cancer type I may not be staged with lymph node evaluation.

Lines 57-59

Multiple factors have been implicated in this unexpected finding, including the aggressive biology of cervical cancer and its propensity to overcome the patient’s immune system and grow primarily through VEGF activation and angiogenesis.

Note.  Please, add the citation.

Lines 135-136

7123 of these women (55%) underwent minimally invasive surgery

Note: Please, do not start the sentence with the number.  F.e.: A total of 7,123 ….

Table 1

I do not understand, why the subjects are divided into weighted and unweighted sections. Especially when this division is not mentioned in the text.

The lenght of follow-up should be shown in both groups in Table 1.

Lines 208-2011

Several factors may be implicated in our study's superior survival outcomes with MIS. First, MIS is associated with decreased postoperative mortality rates at 30 and 90 days, affecting the overall survival of the study cohort. Second, the distinct biology of Type II uterine cancer may make this disease suitable for an MIS approach.

Note: What do you mean that distinct biology of Type II uterine cancer may make this disease suitable for an MIS approach? Do you mean that uterine cancer Type II is suitable for MIS and Type I is not? You probably want to compare with results of MIS in cervical cancer procedures. In this case, it is necessary to rewritten the statement.

Lines 243-247

Endometrial cancer has been categorized into four molecular subgroups (POLE ultramutated, MSI-H, copy number high, and copy number low), each with distinct prognosis and response to treatment. Investigating the connection between tumor molecular classification and surgical approach could be a potential topic for future studies [21].

Note:  An abbreviation should be explained: MSI-H = microsatellite instability -hypermutated.

You can add to this paragraph, which is not however related to your results something as follows: Current guidelines from the National Comprehensive Cancer Network recommend universal testing for mismatch repair (MMR) or MSI status and acknowledge potential benefit to further molecular analysis of tumour protein TP53 and POLE status.  (Abu-Rustum et al. NCCN Guidelines Insights: Uterine Neoplasms, Version 3.2021. J Natl Compr Canc Netw. 2021 Aug 1;19(8):888-895. doi: 10.6004/jnccn.2021.0038. PMID: 34416706.) Or you can delete whole paragraph.

Citations

Citations must be edited and unified. Sometimes the full first name is given, sometimes not. For citation 9, no authors are listed at all. I do not understand quote No. 13 at all. Etc.

Author Response

Responses to Reviewer #3:

General comment by the reviewer:

Dear Editor,

this study, which compares as the main outcome overall survival after MIS and laparotomy operations in women with endometrial carcinoma type II, has two main limitations - as the authors write in the discussion section. First, it is not clear from the database whether the women died as a result of their oncological diseases or other, for example, internal diseases. Second, both groups of women may be very heterogeneous, subjects were not randomized, and nothing is known about their comorbidities, ASA, or performance status. It is possible that patients who were more obese and more ill were selected for laparotomy, and therefore died earlier than women in the MIS group. I leave it to the editor to decide whether to accept the article for publication.

Author response:

We thank the reviewer for their insightful comments. We acknowledge the significant limitation in our study due to the absence of data on cancer-specific survival within the National Cancer Database. Furthermore, the retrospective nature of our study introduces inherent confounding factors. To address these challenges, we employed rigorous statistical methodologies, including multivariable analysis and inverse propensity weighting, aiming to mitigate the impact of confounders. Notably, we incorporated the Charlson comorbidity index into the IPTW model, as demonstrated in Table 1, to assess the effect of comorbidities on survival outcomes. Additionally, our analysis of the trends in minimally invasive surgery (MIS) utilization and its correlation with improved overall survival over time further indicates a potential association between increased MIS use and enhanced survival rates.

Specific comments:

Reviewer comment/question 1:

Lines 44-45. Endometrial cancer is surgically staged with hysterectomy, bilateral salpingo-oophorectomy, and lymph node evaluation. Note: The better statement is: “Endometrial cancer type II is surgically staged with hysterectomy, bilateral salpingo-oophorectomy, and lymph node evaluation.“Low risk endometrial cancer type I may not be staged with lymph node evaluation..

Author response 1:

Modified. Line: 62     

Reviewer comment/question 2:

 Lines 57-59. Multiple factors have been implicated in this unexpected finding, including the aggressive biology of cervical cancer and its propensity to overcome the patient’s immune system and grow primarily through VEGF activation and angiogenesis. Note.  Please, add the citation.

Author response 2:

References added. Line 79

Reviewer comment/question 3:

Lines 135-136. 7123 of these women (55%) underwent minimally invasive surgery. Note: Please, do not start the sentence with the number.  F.e.: A total of 7,123 ….

Author response 3:

Modified

Reviewer comment/question 4:

Table 1. I do not understand, why the subjects are divided into weighted and unweighted

sections. Especially when this division is not mentioned in the text. The lenght of follow-up should be shown in both groups in Table 1.

Author response 4:

Length of follow up is highlighted in Table 1. Designation of unweighed and cohort after IPTW added in the description of Table 1.

We hope that these changes and explanations adequately respond to the reviewer’s comments.

Thank you for your attention.

Round 2

Reviewer 1 Report

Comments and Suggestions for Authors

I appreciated that you took into account the aggressive histological type of the included cases and the absence of information on molecular classification. With the stage assignment based on the updated system of stages IC, IIC and III made in the manuscript and reflected in the tables, the study takes on current significance.

Comments on the Quality of English Language

English is understandable